# Spectral Modification of Graphs
# for Improved Spectral Clustering

**Ioannis Koutis**
Department of Computer Science
New Jersey Institute of Technology
Newark, NJ 07102
ikoutis@njit.edu

**Huong Le**
Department of Computer Science
New Jersey Institute of Technology
Newark, NJ 07102
hyl4@njit.edu

## Abstract

Spectral clustering algorithms provide approximate solutions to hard optimization problems that formulate graph partitioning in terms of the graph conductance. It is well understood that the quality of these approximate solutions is negatively affected by a possibly significant gap between the conductance and the second eigenvalue of the graph. In this paper we show that for any graph $G$, there exists a 'spectral maximizer' graph $H$ which is cut-similar to $G$, but has eigenvalues that are near the theoretical limit implied by the cut structure of $G$. Applying then spectral clustering on $H$ has the potential to produce improved cuts that also exist in $G$ due to the cut similarity. This leads to the second contribution of this work: we describe a practical spectral modification algorithm that raises the eigenvalues of the input graph, while preserving its cuts. Combined with spectral clustering on the modified graph, this yields demonstrably improved cuts.

## 1    Introduction

Spectral Clustering is a widely known family of algorithms that use eigenvectors to partition the vertices of a graph into meaningful clusters. The introduction of spectral partitioning methods goes back to the work of Donath and Hoffman [8] who used eigenvectors for partitioning logic circuits, but owes its popularity to the work of Shi and Malik [25] who brought it in the realm of computer vision and machine learning, subsequently leading to a vast amount of related works. Several other clustering methods have since emerged, including of course methods based on neural networks. But spectral clustering remains a frequently used baseline, and a serious contender to state-of-the-art graph embedding methods, e.g. [20, 11, 28, 22].

The remarkable performance of spectral clustering is possibly due to the fact that it produces outputs with theoretically understood approximation properties. Roughly speaking, spectral clustering computes the second eigenvalue $\lambda$ of the normalized graph Laplacian as an approximation to the graph conductance, i.e. the value of the optimal cut. Cheeger inequality shows that while $\lambda$ is never greater than $\phi$, it can be as small as $\phi^2$ [6]. That implies that the approximation can be a factor of $(\phi/\lambda)$ away from the optimal value, which can be up to $O(n)$ even for unweighted graphs. While this may be often a pessimistic estimate, there are known families of graphs where the estimate is realized; in such graphs, spectral clustering computes cuts that are far from optimal [12]. It is thus understood that the ratio $(\phi/\lambda)$ affects directly the quality of spectral clustering, a fact that is viewed as an inherent limitation.

This paper shows that this limitation can be greatly alleviated via spectral modification: a set of operations that approximately preserve the cut structure of the input while 'raising' its spectrum, in effect suppressing the ratio $(\phi/\lambda)$ and improving the output.

## 2 Spectral Modification: High-level Overview and Context

This section collects a number of required notions from spectral graph theory and puts spectral modification in perspective with important recent discoveries that inspire it. It also describes the motivation for our work and gives a high-level overview that may be useful for the reader before we delve into more technical details.

### 2.1 Cut and Spectral Similarity

Let $G = (V, E, w)$ be a weighted graph. The Laplacian matrix $L_G$ of graph $G$ is defined by: (i) $L_G(u, v) = -w_{uv}$ and (ii) $L_G(u, u) = -\sum_{u \neq v} L_G(u, v)$.

The **quadratic form** of a semi-positive definite matrix $A$ is defined by $\mathcal{R}(A, x) = x^T A x$. For a subset of vertices $S \subseteq V$, we denote by $cut_G(S)$ the total weight of the edges leaving the set $S$.

Let $G$ and $H$ be two weighted graphs. We say that the two graphs are $\rho$-cut similar, if there exist numbers $\alpha, \beta$ with $\rho = \alpha/\beta$, such that for all $S \subset V$, we have $\alpha \cdot cut_H(S) \leq cut_G(S) \leq \beta \cdot cut_H(S)$. We say that the two graphs are $\rho$-spectral similar, if there exist numbers $\alpha, \beta$ with $\rho = \alpha/\beta$ such that for all real vectors $x$, we have $\alpha \cdot \mathcal{R}(L_H, x) \leq \mathcal{R}(L_G, x) \leq \beta \cdot \mathcal{R}(L_H, x)$.

It is well understood that $\rho$-spectral similarity implies $\rho$-cut similarity, but not vice-versa [26].

### 2.2 Low-diameter Cut Approximators and Spectral Maximizers

Let $G = (V, E)$ be the path graph on $n$ vertices, and for the sake of simplicity assume that $n$ is a power of 2. Let $\mathcal{T} = (V \cup I, E)$ be the full binary tree, where $V$ is the set of leaves being in one-to-one correspondence with the path vertices as illustrated in Figure 1a, and $I$ is the set of internal vertices. An interesting feature of $\mathcal{T}$ is that it provides a **cut-approximator** for $G$, i.e. it contains information that allows estimating all cuts in $G$, within a factor of 2. In section 3, we describe how the cut approximator $\mathcal{T}$ gives rise to a weighted complete graph $H = (V, E, w)$ on the original set of vertices $V$, via a canonical process of eliminating the internal vertices of $\mathcal{T}$; figure 1b provides a glimpse to the edge weights of $H$. Graph $H$ is $O(1)$-cut similar with $G$, but with a very different eigenvalue distribution, as illustrated in Figure 1c. More specifically, the second eigenvalue $\lambda$ of the normalized Laplacian of $G$ is $\Theta(1/n^2)$, while that of $H$ is $\Omega(1/(n \log n))$, essentially closing the gap with the conductance $\phi = \Theta(1/n)$. An alternative way of viewing this is that $H$ has a second eigenvalue which –up to an $O(\log n)$ factor– is the maximum possible, since the eigenvalue is always smaller than $\phi$. In some sense, the same is true for all eigenvalues of $H$, which leads us to call $H$ a **spectral maximizer** of $G$. These properties of $H$ can be proved using only the logarithmic diameter of $\mathcal{T}$ and the fact that $\mathcal{T}$ is a cut-approximator.

These observations set the backdrop for the idea of spectral modification, which aims to modify the input graph $G$ in order to bring it spectrally closer to its maximizer $H$. It is worth noting that, in some sense, spectral modification is an objective countering that of spectral graph sparsification, which aims to spectrally preserve a graph [2].

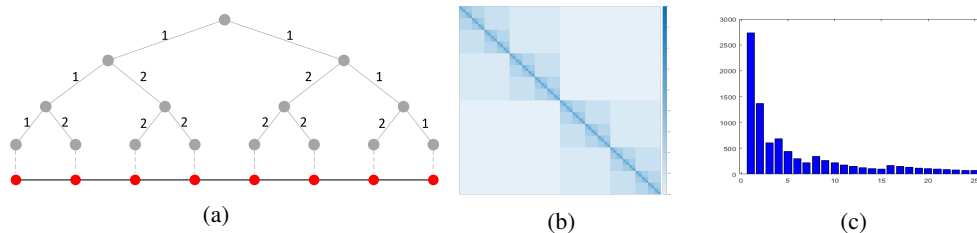

(a)          (b)          (c)

Figure 1: **(a)** The path graph $G$ and the cut-approximating binary tree $\mathcal{R}$. The binary tree is depicted with weights that are discussed in section 3. **(b)** Heatmap of the log-entries of the adjacency matrix for $n = 8196$ of the spectral maximizer $H$. It can be seen that $H$ is a dense graph that inherits the tri-diagonal path structure, but also has other long-range edges. **(c)** Ratios of the first 30 normalized eigenvalues of $H$ and $G$, for $n = 8196$. $H$ has significantly larger eigenvalues.

## 2.3 Contributions and Perspective

A key contribution of this paper is the observation that every graph $G$ has a spectral maximizer $H$. The path-tree example of the previous section is merely an instantiation of our central claim, but a vast generalization is possible (with a small loss), using the fact that all graphs have low-diameter cut approximators, as shown by Räcke [23]. Technically, this is captured by a Cheeger-like inequality that we present in Section 3.3. We show that the inequality applies not only for the standard normalized cuts problem, but also for generalized cut problems that capture semi-supervised clustering problems.

The original result of [23] has undergone several subsequent algorithmic improvements and refinements [3, 14, 24, 19]. It is currently possible to compute a cut approximator in nearly-linear [1] time [19]; this implies a similar time for the construction of a maximizer. As discussed in the previous section, the approximator is a compact representation of **all** cuts in a graph, and thus it is likely that its computation is a waste, when we only want to compute a $k$-clustering. Indeed, all existing algorithms are complicated and far from practical.

On the other hand a significant strength of spectral clustering is its **speed**, due to the existence of provably fast linear system solvers for graph Laplacians [16, 17]. A theoretical upper bound for the computation of $k$ eigenvectors is $O(km \log^2 m)$, where $m$ is the number of edges in the graph; in practice, for a graph with millions of edges, one eigenvector can be computed in mere seconds on standard hardware, without even exploiting the ample potential for parallelism.

This motivates the second contribution of the paper: a **fast** algorithm that modifies the input graph $G$ into a graph $\mathcal{M}$ which is spectrally closer to the maximizer $H$, and thus more amenable to spectral clustering. The emphasis here is in the running time of the modification algorithm and the size of its output. These are kept low in order to not severely impact the speed of spectral clustering. We present the algorithm, and discuss its properties in Section 4.

Finally, applying spectral clustering on graph $\mathcal{M}$ and mapping the output back to $G$ has the potential to 'discover' dramatically different and improved cuts. One such case is illustrated in Figure 2, on a known bad case of spectral clustering taken from [12].

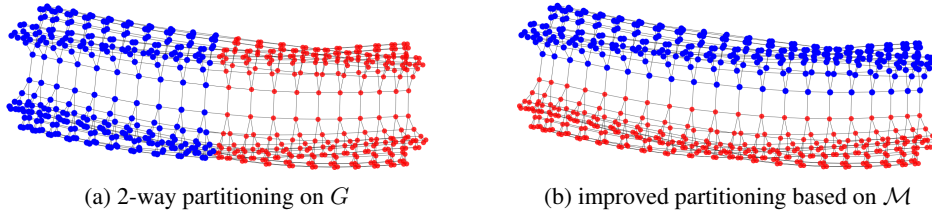

(a) 2-way partitioning on $G$          (b) improved partitioning based on $\mathcal{M}$

Figure 2: Input $G$ is a direct graph product of: (i) the path graph, (ii) a graph consisting of two binary trees with their roots connected [12]. The modification of $G$ **sways** the lowest eigenvector away from the cut computed in $G$. The asymptotic improvement in the value of the cut is $O(n^{1/4})$.

We note that different graph modification ideas have been explored in previous works (e.g. [27, 1, 4]). In particular, in the context of 'regularized spectral clustering', it has been observed that adding a small copy of the identity matrix or the complete graph onto the input graph $G$, improves the quality of spectral clustering [1, 21]. The improved performance has been partially explained for block-stochastic models and stochastic social network graphs [13, 29] . In the latter case, the improvement is attributed to the 'masking' of unbalanced sparse cuts in the graph caused by altering their cut ratio [29]. It is conceivable that the theoretical results of this paper will help shed additional light on regularized spectral clustering. It is clear though, that regularized spectral clustering does not yield improvement such as that in Figure 2.

## 3 Cut Approximators, Spectral Maximizers and Cheeger Inequalities

In this section we prove our main claim that for every graph $G$ there exists another graph $H$ which is cut-similar to $G$ but satisfies a tight Cheeger inequality. We will first state our claims, and give the proofs in subsection 3.4.

## 3.1 Definitions of Graph Objects

**Definition 3.1** (Hierarchical Cut Decomposition). *A hierarchical cut decomposition for a graph $G = (V, E, w)$ is represented as a rooted tree $\mathcal{T} = (V \cup I, E', w')$, with the following properties:*

*(i) Every vertex $u$ of $\mathcal{T}$ identifies a set $S_u \subseteq V$.*
*(ii) If $r$ is the root of $\mathcal{T}$ then $S_r = V$.*
*(iii) If $u$ has children $v_1, \ldots, v_t$ in $\mathcal{T}$. then $S_{v_i} \cap S_{v_j} = \emptyset$ for all $(i, j)$.*
*(iv) If $u$ is the parent of $v$ in $\mathcal{T}$ then $w'(u, v) = cut_G(v)$.*

**Definition 3.2** ($\alpha$-Cut Approximator). *We say that a hierarchical decomposition $\mathcal{T} = (V \cup I, E', w')$ for $G$ is an $\alpha$-cut approximator for $G$ if for all $S \subseteq V$ there exists a set $I_S \subseteq I$ such that*

$$cut_G(S) \leq cut_{\mathcal{T}}(S \cup I_S) \leq \alpha \cdot cut_G(S).$$

Given a graph $G$ and an associated cut approximator $\mathcal{T}$ we now define the **spectral maximizer** for $G$ – the choice of terminology will be justified subsequently.

**Definition 3.3** (Spectral Maximizer). *Let $\mathcal{T} = (V \cup I, E')$ be a cut approximator for a graph $G = (V, E, w)$ and let*

$$L_{\mathcal{T}} = \begin{pmatrix} L_I & V \\ V^T & D \end{pmatrix}$$

*ordered so that its first $|I|$ rows are indexed by $I$ in an arbitrary order, and its last $|V|$ rows are indexed by $V$ in the given order. We define the graph maximizer $H$ to be the graph with Laplacian matrix $L_H = D - V^T L_I^{-1} V$.*

The matrix $D - V^T L_I^{-1} V$ in the above definition is known as the **Schur complement** with respect to the elimination of the vertices/variables in $I$, and given the fact that $L_{\mathcal{T}}$ is Laplacian, it is well known to be a Laplacian matrix (e.g see [9]). Graph theoretically, the elimination of a vertex $v$ from a graph introduces a weighted clique on the neighbors of $v$. The elimination of a set of vertices $I$ can be performed as a sequence of vertex eliminations (in an arbitrary order).

**Important Remark:** We use the term 'spectral maximizer' for brevity and simplicity. It should be made clear that the spectral maximizer is **not** a unique graph, as it depends on $\mathcal{T}$.

## 3.2 Properties of Spectral Maximizers

In order to state our claims we **fix a triple** $(G, \mathcal{T}(\alpha), H)$, where $G$ is a graph, $\mathcal{T}$ is an associated $\alpha$-cut approximator and $H$ is the spectral maximizer corresponding to $\mathcal{T}$. We will also denote by $diam(\mathcal{T})$ the diameter of the tree, i.e. the number of edges on the longest path in $\mathcal{T}$.

We first introduce some required notation additional to that from Section 2.1. Let $G$ and $H$ be two graphs on the same vertex set, with the requirement that $G$ is connected. In particular, $H$ may be not connected, or not even using a susbet of $V$. Then we say that $G$ **spectrally dominates** $H$, if for all vectors $x$ we have $\mathcal{R}(L_G, x) \geq \mathcal{R}(L_H, x)$. We denote spectral domination by $G \succeq H$. We also write $\alpha \cdot G$ to denote the graph $G$ with its weights multiplied by $\alpha$.

**Theorem 3.1** (Spectral Domination of Cut Structure). *Given a triple $(G, \mathcal{T}(\alpha), H)$, let $\tilde{G}$ be an arbitrary graph which is $\rho$-cut similar to $G$. Then, we have $diam(\mathcal{T}) \cdot \rho \cdot H \succeq \tilde{G}$.*

**Theorem 3.2** (Cut Similarity of Spectral Maximizer). *Given a triple $(G, \mathcal{T}(\alpha), H)$, the maximizer $H$ is $\alpha \cdot diam(\mathcal{T})$-cut similar with $G$. In particular, we have $cut_H(S)/\alpha \leq cut_G(S) \leq diam(\mathcal{T})cut_H(S)$.*

We are now ready to discuss the justification for the term 'spectral maximizer'. The reader should think of the parameters $diam(\mathcal{T})$ and $\alpha$ as small, i.e. of size $\tilde{O}(1)$[2]. Then, Theorem 3.1 shows that –up to a $\tilde{O}(1)$ factor– $H$ spectrally dominates **every** graph that is $\tilde{O}(1)$-cut similar with $G$. This directly implies that –up to the same factor– the $i^{th}$ eigenvalue of $L_H$ is greater than that of $L_{\tilde{G}}$, for every graph $\tilde{G}$ which is cut-similar to $G$. Combined with Theorem 3.2, we get that $L_H$ has nearly the maximum possible eigenvalues that **any** graph with similar cuts can have. In the particular case of $\lambda_2$

we show that it is is actually within $\tilde{O}(1)$ from the graph conductance. This extends to a generalized notion of conductance with algorithmic implications for **supervised clustering**; we discuss this in the supplementary material.

### 3.3 Cheeger Inequalities for Spectral Maximizers

**Definition 3.4** (Generalized Conductance). *Let $A$ and $B$ be two graphs on the same set of vertices $V$. We define the generalized conductance $\phi(A, B)$ of the pair as: $\phi(A, B) = \min_{S \subseteq V} \frac{cut_A(S)}{cut_B(S)}$.*

**Definition 3.5** (Second Generalized Eigenvalue). *The smallest generalized eigenvalue of a pair of graphs $(A, B)$ is given by $\lambda_2(A, B) = \min_x \frac{x^T L_A x}{x^T L_B x}$.*

The generalized definition encompasses the standard conductance of a graph. Concretely, let $K$ be the complete weighted graph, where the weight of edge $(u, v)$ is set to be $w_K(u, v) = vol_A(u)vol_A(v)$, i.e. the product of the degrees of $u$ and $v$ in $A$. Also, let $\lambda_2$ denote the second eigenvalue of the normalized Laplacian of $A$, i.e. $\hat{L} = D^{-1/2}L_A D^{-1/2}$, where $D$ is the diagonal matrix of the vertex degrees in $A$. Then, it is easy to show that:

$$\phi(A, K) = \phi(A) = \min_{S \subseteq V} \frac{cut_A(S)}{vol_K(S)vol_K(V-S)} \quad \text{and} \quad \lambda_2(A, K) = \lambda_2.$$

The Cheeger inequality [6] states that $\lambda_2 \geq \phi^2/2$. A Cheeger inequality is also known for the generalized conductance [7]: $\lambda_2(A, B) \geq \phi(A, B)\phi(A)/8$.

We prove the following Theorem.

**Theorem 3.3** (Extended Cheeger Inequality for Cut Structure).
*For any graph $G$, there exists a graph $H$ such that (i) $H$ is $\tilde{O}(1)$-cut similar with $G$, and (ii) $H$ satisfies the following inequality for all graphs $B$:*

$$\lambda_2(H, B) \leq \phi(H, B) \leq \tilde{O}(1)\lambda_2(H, B).$$

A consequence of Theorem 3.3 is that the actual performance of spectral clustering on a given graph $G$ ultimately depends on its 'spectral distance' from its maximizer $H$. This is captured in the following Corollary.

**Corollary 3.1** (Actual Cheeger Inequality).
*Let $G$ be a graph and $H$ be the graph whose existence is guaranteed by Theorem 3.3. Further, suppose that $G$ and $H$ are $\delta$-spectral similar. Then, for all graphs $B$, $G$ satisfies the following inequality: $\lambda_2(G, B) \leq \phi(G, B) \leq \tilde{O}(\delta)\lambda_2(G, B)$.*

### 3.4 Proofs

In this section we simplify the notation and sometimes use $G$ to mean both a graph and its corresponding Laplacian $L_G$.

**Lemma 3.1.** *(Edge-Path Support [5]) Let $P$ be an unweighted path graph on $k$ vertices, with endpoints $u_1, u_k$. Also let $E_{u_1 u_k}$ be the graph consisting only of the edge $(u_1, u_k)$. Then we have $kP \succeq E_{u_1 u_k}$.*

**Lemma 3.2** (Quadratic form of Schur complement). *Let $H$ and $\mathcal{T}$ be the graphs matrices appearing in Definition 3.3. We have*

$$\mathcal{R}(H, x) = \min_{y \in \mathbb{R}^{|I|}} \mathcal{R}(\mathcal{T}, \begin{pmatrix} y \\ x \end{pmatrix}).$$

We finally need the following (adjusted) Lemma from [23, 3]:

**Lemma 3.3.** *Every graph $G$ has an $\tilde{O}(1)$ cut-approximator $\mathcal{R}$. The diameter of $\mathcal{T}$ is $O(\log n)$, where $n$ is the number of vertices in $G$.*

We are now ready to proceed with the proofs.

*Proof.* (of Theorem 3.1) We first show the intermediate claim $diam(\mathcal{T}) \cdot \mathcal{T} \succeq G$. The technique uses elements from support theory [5]. Let $E_{uv}$ be an arbitrary edge of $G$ of weight $w_{uv}$. Let $P_{uv}$ be

the unique path between $u$ and $v$ in $\mathcal{R}$; notice that by definition the path has length at most $diam(\mathcal{T})$. We observe that, by construction of $\mathcal{R}$, we have $\mathcal{T} = \sum_{(u,v) \in G} w_{uv} P_{uv}$. Let $y, x$ be arbitrary vectors of appropriate dimensions, and $z = [y,x]^T$. We have

$$\frac{\mathcal{R}(\mathcal{T},z)}{\mathcal{R}(G,z)} = \frac{\sum_{(u,v)\in G} w_{uv} \mathcal{R}(P_{uv},z)}{\sum_{(u,v)\in G} w_{uv} \mathcal{R}(E_{uv},z)} \geq \min_{(u,v)\in G} \frac{\mathcal{R}(P_{uv},z)}{\mathcal{R}(E_{uv},z)} \geq 1/diam(\mathcal{T}).$$

The first inequality is standard for a ratio of sums of positive numbers, and the second inequality is an application of lemma 3.1. This proves the intermediate claim. Notice now that since the claim holds for all vectors $z = [y,x]^T$ for arbitrary $y$, it also holds for vectors where $y$ is defined as in Lemma 3.2. That implies $\mathcal{T}(H,x) \geq \mathcal{T}(G,x)/diam(\mathcal{T})$, i.e. $diam(\mathcal{T}) \cdot H \succeq G$.

To prove the claim for a $G'$ which is $\rho$-cut similar to $G$, we observe that the above proof can be repeated if we replace $\mathcal{T}$ with $\mathcal{T}' = \sum_{(u,v)\in G} w'_{uv} P_{uv}$. Thus we get $diam(\mathcal{T}) \cdot \mathcal{T}' \succeq G'$ **(A)**. Notice that $\mathcal{T}'$ keeps the same edges of $\mathcal{T}$ but with different weights. Observe now that if $v$ is a vertex in $\mathcal{T}'$ then the edge to its parent has weight equal to $cut_{G'}(S_v)$, where $S_v$ is the set identified by $v$ according to the definition of the cut approximator. However by the cut similarity of $G$ and $G'$ we know that $cut_{G'}(S_v) \geq cut_G S_v / \rho$. It follows that the edges of $\mathcal{T}'$ have weight at most $\rho$ times smaller than their weights in $\mathcal{T}$, which directly implies that $\mathcal{T} \preceq \rho \mathcal{T}'$. Substituting into inequality (A) above, we get that $\rho \cdot diam(\mathcal{T}) \cdot \mathcal{T} \succeq G'$. Then applying lemma 3.2 one more time gives the claim. $\qquad\square$

*Proof.* (of Theorem 3.2) The proof is a relatively easy consequence of lemma 3.2 and definition 3.2. We include it in the supplementary material. $\qquad\square$

*Proof.* (of Theorem 3.3) Let $(G, \mathcal{T}(\alpha), H)$ be the given triple. Also, let $B = (V, E, w)$ be an arbitrary graph. The first part of the inequality is trivial. Let $x$ be the eigenvector corresponding to the smallest non-zero eigenvalue of the generalized problem $L_H x = \lambda L_B x$. Using the standard Courant-Fischer characterization of eigenvalues, we have

$$\lambda_2(H,B) = \frac{\mathcal{R}(L_H,x)}{\mathcal{R}(L_B,x)} = \frac{\mathcal{R}(L_{\mathcal{T}},z)}{\mathcal{R}(L_B,x)}, \tag{1}$$

where $z$ is the extension of $x$ described in lemma 3.2. For an edge $E_{uv}$, let $P_{uv}$ denote the (unique) path connecting $u$ and $v$ in $\mathcal{T}$. Using lemma 3.1, we get:

$$\mathcal{R}(L_B,x) = \sum_{(u,v)\in B} w_{uv}(x_u - x_v)^2 \leq \sum_{(u,v)\in B} w_{uv}\mathcal{R}(L_{P_{uv}},z) = \sum_{(u,v)\in B} \mathcal{R}(w_{uv}L_{P_{uv}},z)$$

Note that we now get the quadratic form of the graph $\mathcal{T}' = \sum_{(u,v)\in B} w_{uv} P_{uv}$. Because $\mathcal{T}'$ is a sum of paths on $\mathcal{T}$, it has the same edges with $T$. Denote by $w_{\mathcal{T}}(q,q')$ the weight of the edge $(q,q')$ on $\mathcal{T}$, where $q'$ is the parent of $q$. Continuing then on inequality 1, we get

$$\lambda_2(H,B) \geq \frac{\mathcal{R}(L_{\mathcal{T}},z)}{\mathcal{R}(L_{\mathcal{T}'},z)} = \frac{\sum_{(q,q')\in\mathcal{T}} w_{\mathcal{T}}(q,q')(z_q - z_{q'})^2}{\sum_{(q,q')\in\mathcal{T}} w_{\mathcal{T}'}(q,q')(z_q - z_{q'})^2} \geq \min_{q\in\mathcal{T}} \frac{w_{\mathcal{T}}(q,q')}{w_{\mathcal{T}'}(q,q')} \tag{2}$$

If $S_q \subseteq V$ is the set identified by $q$, we have

$$w_{\mathcal{T}}(q,q) = cut_G(S_q) \geq cut_H(S_q)/\alpha,$$

where the inequality comes from Theorem 3.2. Observe now that $(q,q')$ appears on $\mathcal{T}'$ exactly on the paths $P_{uv}$ such $u \in S_q$ and $v \in S'_q$. It follows that the edge $(q,q')$ receives in $\mathcal{T}'$ a total weight equal to the total weight of the edges leaving $S_q$ on $B$, i.e. $w_{\mathcal{T}'}(q,q') = cut_B(S_q)$. Further continuing on inequality 2, we get that

$$\lambda_2(H,B) \geq \min_{q\in\mathcal{T}} \frac{w_{\mathcal{T}}(q,q')}{w_{\mathcal{T}'}(q,q')} \geq \min_q \frac{cut_H(S_q)}{\alpha \cdot cut_B(S_q)} \geq \min_S \frac{cut_H(S)}{\alpha \cdot cut_B(S)} = \phi(H,B)/\alpha.$$

The Theorem then follows by invoking lemma 3.3 and Theorem 3.2. $\qquad\square$

# 4 A Spectral Modification Algorithm

The goal of spectral modification is to construct a **modifier** $M$ of the input graph $G = (V, E, w)$, which is spectrally similar to the maximizer described in Section 3. Then Corollary 3.1 shows that improved Cheeger inequalities also hold for $M$, up to the spectral similarity factor. Echoing the construction of the maximizer in Section 3, we will construct a graph $\mathcal{M}$ on a set of vertices $V \cup V_{add}$, where $V_{add}$ is a set of additional vertices. The modifier $M$ is then defined as the Schur complement of $\mathcal{M}$ with respect to the elimination of the nodes in $V_{add}$. We solve the generalized eigenvalue problem $L_M x = \lambda D x$, where $D$ is the diagonal of $L_G$. The modifier $M$ is a dense graph, but we effectively use only $\mathcal{M}$. We accomplish that using standard techniques that we discuss in the supplementary file.

**Cut Approximators for Trees.** Towards designing a modification algorithm, we observe that computing a low-diameter cut approximator of a tree $T$ is can be carried out with a recursive top-down analysis of the cut structure of $T$, in $O(n \log n)$ time, essentially following the algorithm in [23]; key to the algorithm is a linear time algorithm for computing the sparsest cut on a tree. A low-diameter cut approximator for a tree can also be constructed in a bottom-up fashion in $O(n)$ time, using the decompositions from [15]. Our code implements the linear time algorithm.

We consider the following general framework for spectral modification. Given a graph $G = (V, E, w)$:

**(a)** Compute a set of weighted trees $T_1, \ldots, T_k$ on vertex set $V$. **[tree decomposition step]**
**(b)** Compute a cut approximator $\mathcal{M}_j$ for each tree $T_j$.
**(c)** Form the graph $\mathcal{M} = \alpha G + \mathcal{M}_1 + \ldots \mathcal{M}_k$.

The cut approximators $T_j$ in step (b) share the same set of leaves $V$, but each $T_j$ has its own set of additional internal vertices $V_{add_j}$. Thus, the weighted graphs in the sum of step (c) have mutually disjoint edge sets, and the sum simply denotes the union of all these edges. The vertex set of $\mathcal{M}$ is $V \cup V_{add}$, where $V_{add} = \bigcup_j V_{add_j}$.

**Tree Decomposition Step.** In this step we aim to process the input graph $G$ in order to compute a set of trees, such that the sum of their maximizers is spectrally close to the maximizer of $G$. There exist several potential ways to perform that. We now give an algebra-based heuristic algorithm that we have implemented and used in our experiments.

```
1: procedure ENERGY_TD(G, k)
2:     z ← approximate second eigenvector of L_G x = λDx          ▷ D is the diagonal of L_G
3:     G' ← (V, E, w'), where w'_uv = w_uv(z_u − z_v)²
4:     for j = 1 : k do
5:         R_j = (V, E_j, w') ← maximum weight spanning tree of G'
6:         T_j ← (V, E_j, w)                      ▷ Tree with edge set E_j with weights from G
7:         For each e ∈ E_j, let w'_e = w'_e/df              ▷ Update weights in G'
8:     end for
9:     return {T_1, ..., T_k}
```

ENERGY_TD is based on the following reasoning. Assuming that the graph $G$ is spectrally away from its maximizer $H$, we expect the second eigenvector $z$ to be "bad" in the sense that the associated Rayleigh quotient $\mathcal{R}(G, z)/z^T D z$ is significantly lower than it would have been for the maximizer $H$. Steps 4-7 find $k$ trees in $G$ that yield most of the 'energy' $\mathcal{R}(G, z)$. Adding the maximizers of these trees attempts to directly 'push' the Rayleigh quotient for $z$ higher in the spectrum of the modified graph $M$. At the same time, because the trees $T_j$ are subtrees of $G$, and their maximizers have similar cuts, the modifier $M$ has cuts similar to those in $G$. We further discuss some properties of ENERGY_TD and its running time, in the supplementary file.

## 4.1 Implementation and Experiments

We provide a `MATLAB` implentation. We plan to provide a `Python` implementation in the near future. The submitted code and all future updates can be found in: https://github.com/ikoutis/spectral-modification

**Remark on Baseline Spectral Clustering:** We use the baseline spectral clustering implementation from [7]. We solve the eigenvalue problem $L_G x = \lambda D x$, which yields the standard embedding. A differentiation is that we further process the embedding by projecting the points onto the unit hypersphere, as analyzed in [18]. This actually yields a significant improvement of the baseline.

**Parameter Settings:** For all our experiments we set $k = 3$, $df = 1/2$, and $\alpha = 1$ in ENERGY_TD.

**Synthetic Datasets.** The synthetic example described in Figure 2 highlights the potential of spectral modification to induce the computation of asymptotically better cuts in graphs with 'elongated' features, or high diameter. The output has been computationally verified for a range of values for $n$ (up to millions). In the supplementary file we also describe a synthetic example of a weighted where spectral modification yields a cut smaller by a $\Theta(1/n)$ factor. In Figure 3, we also give a synthetic example taken from [7], where spectral modification clearly outperforms even a supervised method.

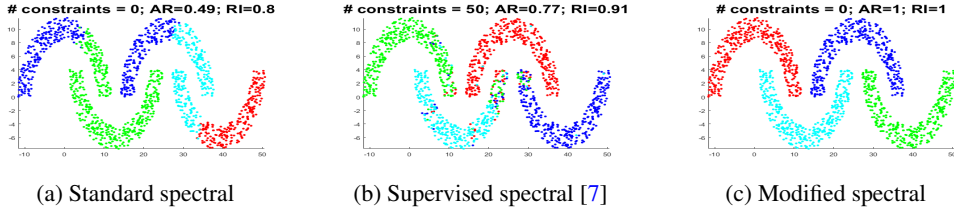

(a) Standard spectral       (b) Supervised spectral [7]       (c) Modified spectral

Figure 3: **(a)** The '4-moons' example from [7]. (A)RI is the Adjusted Rand Index.

**Social Networks.** We performed experiments with four graphs (BlogCatalog, PPI, Wikipedia, Flickr) used as a benchmark in the recent literature [20, 22]. We compare against NetMF [22] as it has previously reported an improvement over DeepWalk [20] and other competing methods. The evaluation methodology is identical to that in [22]. The second normalized eigenvalue $\lambda$ of these graphs are quite high (0.43,0.49, 0.20, 0.06 respectively) and so there is little room for improvement. Nevertheless we observe improvements in the standard Micro-F1 scores. We cannot however attribute them directly to our theory, as it is not sensitive to $\tilde{O}(1)$ factors. The dimension of the embedding is equal to the number of clusters, except for the Flickr data set which is set to 128 because NetMF method is too expensive to be run on dimension 195 (# clusters). We also wish to highlight the fact that the implemented version of baseline spectral clustering performs much better than standard version. A more detailed discussion can be found in the supplementary file.

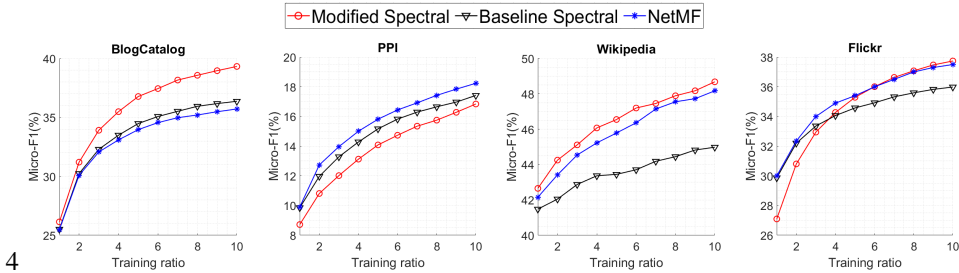

Figure 4: Micro-F1 scores in 10x cross-validation using LIBLINEAR [10].

**Conclusion**. The performance of spectral clustering depends crucially on spectral properties of its input graph, which most often force it to output clusters of poor approximation quality. This has been viewed as an inherent limitation of spectral clustering. We show however that for any input graph, there exists a 'maximizer' graph with similar cuts, but with an eigenvalue distribution which is favorable for spectral clustering. We propose a spectral modification algorithm that attempts to exploit this fact via fast operations that improve the eigenvalue distribution of the input without changing its cut structure. The implemented spectral modification algorithm is heuristic and subject to various improvements. Nevertheless, it yields demonstrable asymptotic improvements in a number of adversarial instances. In future work we will explore the performance of spectral modification on larger and more diverse sets of instances, and the implementation of modification algorithms with theoretical guarantees

**Acknowledgements.** This work has been partially supported by grants CCF-1149048, CCF-1813374.

## Footnotes

[1] $O(m \log^c m)$ time, where $m$ is the number of edges in $G$ and $c$ is a fairly large constant.

[2]We use the $\tilde{O}(\cdot)$ notation to hide factors logarithmic in $n$, that we do not attempt to optimize.

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
