[Supplementary Material · final_2695_with_supplementary.pdf]

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

| (a) Standard spectral | (b) Supervised spectral [6] | (c) Modified spectral |

Figure 3: **(a)** The '4-moons' example from [6]. (A)RI is the Adjusted Rand Index.

**Social Networks.** We performed experiments with four graphs (BlogCatalog, PPI, Wikipedia, Flickr) used as a benchmark in the recent literature [23, 25]. We compare against NetMF [25] as it has previously reported an improvement over DeepWalk [23] and other competing methods. The evaluation methodology is identical to that in [25]. The second normalized eigenvalue $\lambda$ of these graphs are quite high (0.43,0.49, 0.20, 0.06 respectively) and so there is little room for improvement. Nevertheless we observe improvements in the standard Micro-F1 scores. We cannot however attribute them directly to our theory, as it is not sensitive to $\tilde{O}(1)$ factors. The dimension of the embedding is equal to the number of clusters, except for the Flickr data set which is set to 128 because NetMF method is too expensive to be run on dimension 195 (# clusters). We also wish to highlight the fact that the implemented version of baseline spectral clustering performs much better than standard version. A more detailed discussion is given in the supplementary file.

Figure 4: Micro-F1 scores in 10x cross-validation using LIBLINEAR [9].

**Conclusion**. The performance of spectral clustering depends crucially on spectral properties of its input graph, which most often force it to output clusters of poor approximation quality. This has been viewed as an inherent limitation of spectral clustering. We show however that for any input graph, there exists a 'maximizer' graph with similar cuts, but with an eigenvalue distribution which is favorable for spectral clustering. We propose a spectral modification algorithm that attempts to exploit this fact via fast operations that improve the eigenvalue distribution of the input without changing its cut structure. The implemented spectral modification algorithm is heuristic and subject to various improvements. Nevertheless, it yields demonstrable asymptotic improvements in a number of adversarial instances. In future work we will explore the performance of spectral modification on larger and more diverse sets of instances, and the implementation of modification algorithms with theoretical guarantees

**Acknowledgements.** This work has been partially supported by grants CCF-1149048, CCF-1813374.

## Footnotes

[1] $O(m \log^c m)$ time, where $m$ is the number of edges in $G$ and $c$ is a fairly large constant.

[2]We use the $\tilde{O}(\cdot)$ notation to hide factors logarithmic in $n$, that we do not attempt to optimize.

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

# 5 Supplementary Algorithms and Theory

## 5.1 Missing Proofs

*Proof.* (of Theorem 3.2) From Theorem 3.1 we get directly that $cut_S(G) \leq diam(\mathcal{T})cut_S(H)$ **(A)** for all sets $S \subseteq V$. Now, let us fix some $S \subseteq V$. We define $x_S$ to be the indicator vector of $S$ with: $x_S(v) = 1$ is $v \in S$ and $x_S(v) = 0$ otherwise. For any graph $G$, we have

$$\mathcal{R}(G, x_S) = \sum_{u \in S, v \notin S} w_{uv}(x_u - x_v)^2 = cut_G(S). \tag{3}$$

Definition 3.2 identifies a set $I_S \subset I$ of internal nodes of $\mathcal{R}$. We thus construct a vector $z = [x_S, y_{I_S}]^T$, where $y_{I_S}$ is the indicator vector for $I_S$. We then get:

$$cut_H(S) = \mathcal{R}(H, x_S) \leq \mathcal{R}(\mathcal{T}, z) = cut_\mathcal{T}(S \cup I_S) \leq \alpha \cdot cut_G(S) \quad \textbf{(B)}$$

where the first inequality comes from lemma 3.2 and the last one from definition 3.2. Combining (A) and (B) we get that $cut_H(S)/\alpha \leq cut_G(S) \leq diam(\mathcal{T})cut_H(S)$. □

## 5.2 Other Algorithmic Consequences

Spectral clustering is generally viewed as an **unsupervised** method. However it has been suggested that it can also work as a **supervised** algorithm via computing generalized eigenvectors [6]. Corollary 3.1 shows that the theoretical performance of this kind of supervised spectral clustering has the potential to be much more accurate than predicted by the generalized inequality of [6] if the input graph is spectrally close to the maximizer of its cut structure.

Corollary 3.1 also has consequences for classical algorithmic problems. For instance, the isoperimetric number of a graph $G$ is often defined as

$$h = \min_{S \subseteq V} cut_G(S)/(|S| \cdot |V - S|).$$

If we let $B$ to be the complete unweighted graph then $h = \phi(G, B)$. The isoperimetric number has a weaker Cheeger inequality, namely $\lambda_2(L_G) \geq h^2/(2d_{\max})$, where $d_{\max}$ is the maximum degree of the graph. Then inequality 3.1 applies directly and gives a different and usually stronger estimate. A similarly interesting inequality follows for the minimum $s$-$t$ cut problem, if we set $B$ to be the graph consisting only of the $(s, t)$ edge.

## 5.3 Eigenvsolver for modified graphs.

The modifier $M$ is a dense graph, and we effectively use only $\mathcal{M}$. We accomplish this using standard techniques that enable us to effectively use only $\mathcal{M}$: As observed by Spielman and Teng [29], a nearly-linear time implementation of the required eigenvector computation via inverse power methods, requires only solving linear systems of the form $L_M x = b$. In turn, this can be done via solving linear systems of the form $L_\mathcal{M} x' = b'$, as $M$ is simply the product of Gaussian elimination on $\mathcal{M}$ [21]. More specifically, we set $b'$ to agree with $b$ on $V$, and be equal to 0 on the $V_{add}$. Then $x$ is recovered from the $V$ coordinates of $x'$. Solving linear systems on $L_\mathcal{M}$ can be done in time $O(m \log m)$, where $m$ is the number of edges in $\mathcal{M}$, using a fast Laplacian solver [17, 19]. In practice, we use the Combinatorial Multigrid (CMG) solver [18]. The worst-case time required for computing the $k$ vectors used in the embedding is at most $O(km \log^2 m)$ when a standard inverse power method is employed, and assuming that the running time of the linear system solver is $O(m \log m)$. In practice the code is much faster due to the faster than worst-case performance of the linear system solver, and to the used preconditioned eigensolver `lobpcg` [15].

# 6 Supplementary Experiments

## 6.1 A weighted synthetic example

(a) 2-way partitioning on $G$      (b) 2-way partitioning on $\mathcal{M}$

Figure 5: A weighted example. Input $G$ consists of two unit-weight cycle graphs of length $n$, with their corresponding vertices connected by edges of weight $100/n^2$ [10]. Standard spectral clustering cuts 4 unit edges but modified spectral clustering cuts $n$ edges of weight $100/n^2$.

## 6.2 Experiments with social network data

We use 4 labeled datasets that have been widely used as benchmarks [23]. Table 1 summarizes their features.

| Dataset | BlogCatalog | PPI | Wikipedia | Flickr |
|---|---|---|---|---|
| $\|V\|$ | 10,312 | 3,890 | 4,777 | 80,513 |
| $\|E\|$ | 333,983 | 76,584 | 184,812 | 5,899,882 |
| # Labels | 39 | 50 | 40 | 195 |
| $\lambda_2$ | 0.4961 | 0.4316 | 0.2001 | 0.0589 |

Table 1: Dataset features including second eigenvalue $\lambda_2$

Note that the second eigenvalue $\lambda_2$ of these networks is quite high, and recall that the theory we developed is insensitive to $\tilde{O}(1)$ factors. Nevertheless we experimented with the implemented version of spectral modification, following exactly the methodology of [25]: we first compute the embedding and then perform a 10x cross-validation using LIBLINEAR [9], at various levels of supervision, for the standard Micro-F1 and Macro-F1 metrics. We compare against:

- The baseline spectral clustering method, implemented with the radial projection step proposed and analyzed in [20]. The step projects the points onto the unit hypersphere the points computed by the standard embedding.
- The NetMF network embedding method [25] which has been shown to perform better than other recent network embedding methods (e.g. DeepWalk [23], LINE [30]).

The dimension of the embedding is set to be the number of clusters, except for the Flickr dataset where the dimension is set to 128 (reported in [25]), as we were not able to run NetMF on dimension 195 on standard hardware.

Figures 6 and 7 summarize the experiments. Although the numbers are not reported here, we wish to highlight the observation that baseline spectral clustering with radial projection performs significantly better than the standard version that was used in previous works.

Figure 6: Micro-F1 performance for classification using LIBLINEAR [9] (10x cross validation)

Figure 7: Macro-F1 performance for classification using LIBLINEAR [9] (10x cross validation)