[Reviews · NeurIPS 2019]

Reviewer 1



Summary: The paper proposes to modify graphs in order to improve the quality of spectral partitioning on them. The quality is governed by the graph spectrum, and the proposed approach is to change the spectrum while approximately preserving cut values. This is achieved in two steps: first use Racke's embedding of the graph into a cut-preserving tree with extra nodes (this step apparently improves the spectrum), then remove the extra nodes by computing the graph Schur complement, an operation which fully preserves the spectrum including the cut values. The resulting graph has approximately the same cut values and improved spectrum. Since both operations are too costly to be considered practical, the paper proceeds to suggest a heuristic variant of the method. Finally several small experimental visualizations are shown to display the advantage of the proposed method. Evaluation: Ultimately the main idea paper is a direct "black-box" invocation of two known techniques (Racke's tree cut sparsifiers and vertex sparsification by Schur complementation), but it seems potentially useful, and the examples provided in the paper give a clear and crisp proof-of-concept for its advantage. The main weakness of the paper is the lack of empirical evaluation. Given that the paper concedes that the theoretical variant of the method is impractical, and suggests an alternative heuristic version, it seems unfortunate that there is no principled experimental evaluation on even moderate-sized graphs. The few examples given in the paper are small, qualitative and somewhat arbitrary (as opposed to using standard benchmark input graph, comparison to baselines and other methods, principled choice of parameters, numerical evaluation of performance, etc). Put simply, it is not quite clear what to make of an algorithm that has neither firm theoretical guarantees nor certified empirical advantage. In conclusion, the paper suggests a nice and simple way to enhance the quality of spectral partitioning and makes a case for its advantage, then discusses practical considerations, but stops short of substantiating the empirical usefulness of its method. Update post-rebuttal: I thank the authors for their detailed response. I do not find it adequately addresses the concerns raised about empirical evaluation. It argues that their theoretical analysis renders thorough experiments unnecessary; however, as previously noted, their primary algorithm is at present supported by neither proofs (only intuition) nor experiments (only proof-of-concept). As a result, I maintain my score.

Reviewer 2



Overall, I feel the paper is written in a rush and there are many problems in notations and presentation. More importantly, there seem to be a few flaws or statements without sufficient justification, which makes the paper theoretically unsolid. • \phi is used in the introduction, but it is not defined until section 2 • Could you please use [] instead of () for references? The formula are also indexed by (), which is confusing. • Definition 3.2 should be revised a bit. It has two “for all $S \subset V$” • Definition 3.3 should also be slightly revised: there are grammar mistakes and you never specify which quantity is the maximizer, though I guess it should be H. • Since the spectral maximizer depends on the cut approximator, so I think it should be defined to be \alpha-spectral maximizer” with a corresponding parameter \alpha, correct? But seems H is never associated with such a parameter. Please clarify. • What do mean by “not even using a subset of V” while requiring “two graphs on the same vertex set” on line 131 • Theorem 3.2 is confusing. What is S? • Section 3.3 is titled as “Cheeger Inequalities for Spectral Maximizers”. But in theorem 3.3, you only show the existence of such H, but there is no statement whether the H is the spectral maximizer of G. And the Corollary 3.1 depends on Theorem 3.3. So at the end, I don’t see the connection between the results in Section 3.3 and the spectral maximizer in Section 3.2 • It is discussed at the end of page 4, that we should assume the diameter of T to be \tilde{O}(1). I don’t see a rigorous justification of this statement, unless I missed something. • The algorithm does not have theoretical guarantee for claimed properties. The authors addressed most of my questions. However, the correspondence between the existing H and the spectral maximizer is still not clear to me. Perhaps after the authors add clarification, this can be resolved. The theoretical guarantee for the algorithm is not available still, which seems to me is important. Given the current result, I will change my evaluation to 6.

Reviewer 3



In this work the authors study the following problem: given an input graph G, define as Cut(G) the set of all graphs with the same cut structure. The goal is to find a graph H within Cut(G) that dominates spectrally G. The idea is that applying spectral clustering on H will result in better outputs, and will avoid the pathological performance of spectral clustering which actually occurs in real-world datasets. The author(s) prove that Racke's trees serve this purpose. Interestingly, sparse spectral maximizers also exist; just sparsify if needed the maximizer. The algorithm is simple to implement, and can potentially have broader impact. The weakest part in this work are the experiments. It would be interesting to demonstrate better the impact of the method.

[Author Response · NeurIPS 2019]

We thank the reviewers for their nice and helpful comments. We are happy that the reviewers see value and potential in
our main claim that spectral modification can circumvent a fundamental weakness of spectral clustering.

Our modification algorithm is intended as a proof of concept of what we view as a general framework. It is by design
very fast, and it always outputs a graph that spectrally dominates the input graph while preserving its cuts. It is indeed
heuristic in the sense that the output is not provably spectrally close to the maximimizer. However its derivation is not
ad-hoc; it is based on theoretical intuition that we briefly discuss in page 7. In fact, we believe that a still-practical
variant of the algorithm should be analyzable.

There have been several recent works on graph embedding methods. These are quite successful, but appear to lack
any theoretical justification of their empirical performance gains. These methods effectively compute modified linear
operators from the input graph via 'deep walks' that associate nodes at longer ranges, followed by eigenvector-based
embeddings (e.g. see [22]). This is very much in the spirit of our work and we believe that our findings can shed
theoretical light to their success. With this work we want to lay the theoretical groundwork towards that direction,
which we believe merits a separate treatment, as we explain below.

**Comments on the choice of experiments.**

• Our experimental examples are well known and appreciated in spectral graph theory. They are understood to capture a
fundamental weakness of spectral partitioning, and for that reason we would not consider them as "somewhat arbitrary".
We can report that the unsupervised versions of recent graph embedding methods (e.g. NetMF [22]) **fail** to compute
the correct solutions on these examples . Getting the correct output requires a higher dimension in the embedding,
**and** a significant amount of supervision, whereas our result uses no supervision; we will add these experiments in the
supplementary material and arXiv version. So, up to our knowledge, there is no other graph embedding algorithm that
correctly computes a good solution for these difficult instances.

• The **size** of the graphs we use for the visualization had to be small for the visualization to work properly. The same
experiments have been repeated for much larger versions of the same graphs (with millions of nodes), with the same
outcome. That is why Figures (2) and (3) refer to an **asymptotic** improvement (with respect to $n$) of the value of the
cut. Notably, this improvement is with respect to a precise optimization problem (i.e. conductance), and the obtained
solution is optimal. So, with the exception of the image examples, we would not consider these results as "qualitative".
Clearly, demonstrating asymptotic improvements cannot be done via empirical evaluations on fixed-size networks.

• We are aware of the fact that recent graph embedding works provide extensive experimental evaluations. However, in
the absence of any theoretical justification, extensive evaluation is necessary to demonstrate their value; we believe
that this is not the case with our work. Importantly, these evaluations are empirical, i.e. against given 'ground truths'
and not with respect to well-defined optimization problems as in our case. They pertain to the **practical** utility of the
algorithms, which is a markedly distinct topic relative to the more theoretical orientation of our work.

• We do however believe in the applicability of spectral modification at least for certain classes of problems, and
we plan to conduct comparisons with other methods. It should be noted though that these recent works have been
successful mostly in the **supervised** setting. Our algorithm has a supervised version –implied by Theorem 3.3, where an
appropriately formulated graph $B$ can encode supervision information. This supervised extension is also a distinct task
with a host of technical issues that require a separate focus in order to attain the full potential of our approach. In this
context, we plan to also study if and to what extent other embedding methods are realizations of spectral modification.

**Answer to specific comments by reviewer #2.**

• In Theorem 3.3 we wrote "there exists a graph $H$..." in order make the theorem self-contained, without reference to
the preceding definitions. But $H$ is indeed the spectral maximizer; a clarification will be added. Note that Theorem 3.1
essentially states that the maximizer has the maximum possible eigenvalues for the given cuts, but it does not quantify
that maximum. With Theorem 3.1 alone, the second eigenvalue can still be lower than the conductance. Theorem 3.3
states that the second eigenvalue is actually within $\tilde{O}(1)$ from $\phi$, which produces a tight gap in the Cheeger inequality.
This is in fact explained in lines 144-147, but we realize that a clearer explanation is needed.

• The rigorous justification about the diameter of the tree is given in Lemma 3.3, which states that it is $O(\log n)$.

• **Minor Points:** The quality of the spectral maximizer indeed depends on $\alpha$ and the diameter of the tree. We omit them
from the definition for the sake of conceptual simplicity and brevity, as we remark in lines 124-125. We will add symbol $\phi$
in line 25, right after 'conductance'. Line 131 appears problematic due to the fact that the two paragraphs in lines 127–129
and 130–134 were unintentionally swapped. This was just meant as a minor technical point that spectral dominance does not
require $H$ to be connected. We will fix it. In Theorem 3.2 we will add "for all $S \subseteq V$", which specifies what $S$ is. We will
fix the typo in Definition 3.2. In Definition 3.3 we will add a concrete statement that $H$ is the maximizer. [ ] will be used for references.
53

[Meta-Review · NeurIPS 2019]

This paper presents an interesting approach of finding cut-similar graphs that have more desirable spectral properties if one to were, e.g., run spectral clustering. All of the reviewers found some very positive aspects of the paper theoretically. On the other hand, the experiments are considerably underwhelming and the authors did not make much of an effort to address this with the author feedback. Since there is a really nice contribution here, I am recommending this paper for acceptance. However, I would really like to encourage the authors to add more meaningful numerical experiments for a camera version. The current experiments in Section 4.1 can be removed and replaced with some of the experiments suggested by, e.g., Reviewer 3 on some moderately sized (maybe ~100,000 edges), more heterogeneous graphs.